# Two New Mitogenomes of Bibionidae and Their Comparison within the Infraorder Bibionomorpha (Diptera)

**DOI:** 10.3390/genes14071485

**Published:** 2023-07-21

**Authors:** Mei-Ling Xiao, Huan Yuan, Ting-Jing Li, Bin Chen

**Affiliations:** Key Laboratory of Vector Insects, Institute of Entomology and Molecular Biology, Chongqing Normal University, Chongqing 401331, China; 2020110513042@stu.cqnu.edu.cn (M.-L.X.); 2022010513007@stu.cqnu.edu.cn (H.Y.); ltjing1979@hotmail.com (T.-J.L.)

**Keywords:** Bibionomorpha, mitogenomes, mitogenome characteristics, gene rearrangement, phylogenetics, evolution

## Abstract

Despite the worldwide distribution and rich diversity of the infraorder Bibionomorpha in Diptera, the characteristics of mitochondrial genomes (mitogenomes) are still little-known, and the phylogenetics and evolution of the infraorder remains controversial. In the present study, we report complete and annotated mitogenome sequences of *Penthetria simplioipes* and *Plecia hardyi* representing Bibionidae. This is the first report of the complete mitogenomes for the superfamily Bibionoidea. There are 37 genes in each of the complete mitogenomes of all 20 studied species from eight families of four superfamilies within infraorder Bibionomorpha. The Ka/Ks analysis suggests that all 13 PCGs have undergone purifying selection. The gene rearrangement events exist in some families (Keroplatidae, Sciaridae, and Cecidomyiidae) but not in Mycetophilidae in Sciaroidea and also in Scatopsoidea, Anisopodoidea, and Bibionoidea, which suggests that these rearrangement events are derived in the late period in the evolution of the Bibionomorpha. The phylogenetic analysis suggests the phylogenetic relationships of Scatopsoidea + (Anisopodoidea + (Bibionoidea + Sciaroidea)) in Bibionomorpha. The divergence time analysis suggests that Bibionomorpha originated in the Triassic, Scatopsoidea and Anisopodoidea in the late Triassic, Bibionoidea in the Jurassic, and Sciaroidea in the Jurassic to the Cretaceous. The work lays a base for the study of mitogenomes in Bibionomorpha but further work and broader taxon sampling are necessary for a better understanding of the phylogenetics and evolution of the infraorder.

## 1. Introduction

The infraorder Bibionomorpha is one of the largest groups in Diptera with approximately 16,500 known species worldwide [1]. The insects in the infraorder are mostly tiny flies with a slender dark-colored body, long legs, and simple wings. Species in the infraorder are widely distributed across the world, and usually aggregate under moist shady areas, including leaves, grass clippings, moss, and manure. The feeding habits of their larvae are diverse, including detritophagy, saprophagy, predatory, mycophagy, and phytophagy [2]. Some larvae of bibionids and sciarids are found mostly in decaying organic materials such as forest litter, manure, and humus-rich soils, and play an important role as decomposers. Nevertheless, some species are known as economically significant pests, which cause damage to the roots of cultivated plants. The larvae of *Bradysia* cause serious damage to onions, carrots, and edible mushrooms [3], and their adults serve as vectors for the pathogen *Fusarium foetens* on the ornamental plant begonias, causing serious economic loss [4,5].

The classification of the Bibionomorpha has been studied for almost 200 years, and its monophyly has been confirmed by morphological and molecular studies [6,7,8]. The earliest comprehensive classification of the infraorder included 10 families primarily based on their adult character, Perissommatidae, Pachyneuridae, Axymyiidae, Anisopodidae, Canthyloscelidae, Scatopsidae, Cecidomyiidae, Sciaridae, Mycetophilidae sensu lato, and Bibionidae, into Bibionomorpha [6,9]. Nowadays, the infraorder is widely recognized to include four superfamilies (Sciaroidea, Bibionoidea, Anisopodoidea, and Scatopsoidea), 34 families, and almost 1400 genera [1,10].

A comprehensive phylogeny study of Diptera based on 13 protein-coding gene (*PCG*) and two ribosomal gene (*18S*, *28S*) sequences using Bayesian inference (BI) suggested that the family Anisopodidae was not included in the Bibionomorpha [11]. This result is inconsistent with that of the phylogenetic analyses based on morphological data [9]. The phylogenetic analysis of the infraorder based on three nuclear (*18S*, *28S*, and *CAD*) and three mitochondrial (*12S*, *16S*, and *COXI*) genes using maximum likelihood (ML) and BI demonstrated that Bibionoidea and Sciaroidea were monophyletic, the Bibionidae was paraphyletic, Bibionidae and Pachyneuridae were sister groups, and Sciaroidea included Cecidomyiidae [10]. However, earlier studies of Diptera by Oosterbroek et al. demonstrated that Bibionidae was monophyletic, which is consistent with that of phylogenetic analysis based on morphological data [12], and Sciaroidea was paraphyletic based on ribosomal (*28S*) and protein-coding (*CAD*, *TPI*, and *PGD*) genes [10]. The sister relationship of Bibionidae and Keroplatidae was supported by a comprehensive morphological character matrix [13], but Mantič et al. proposed a sister group to the Keroplatidae is a clade containing all the other groups of Sciaroidea, except Ditomyiidae and Cecidomyiidae [14]. Cecidomyiidae was monophyletic without any doubt [7,15,16]. Cecidomyiidae was the sister group to Sciaroidea based on morphology [2,17,18], Cecidomyiidae + Sciaridae was the sister group to Mycetophilidae based on the character of the larva [7], and Cecidomyiidae was the sister group of the Mycetophilidae + Sciaridae based on the morphology character matrix [19].

Mitochondria are vital organelles in eukaryotic cells, which are involved in oxidative phosphorylation [20]. A fast-increasing number of mitogenomes of insect species have been sequenced and annotated since 2004. In 2009, two complete mitogenomes were published for the first time in the infraorder Bibionomorpha (*Mayetiola destructor* and *Rhopalomyia pomum*) [21]. Up to date, complete mitogenome sequences have been reported from three superfamilies; however, there is no complete mitogenome sequence to be reported for the superfamily Bibionoidea in GenBank in the infraorder. The mitogenomes reported in the infraorder are typical circular molecules with the length ranging from 14 kb to 16 kb, and each contains a conserved set of 37 genes, including 13 *PCGs*, large and small ribosomal genes (*rrnL* and *rrnS*), 22 transfer RNAs (*tRNA*) genes, and a control region (*CR*) [22,23,24]. There are complicated gene rearrangements to be identified, e.g., inversion and transposition of tRNA*s* and PCG*s* in Cecidomyiidae [20], a *trnE* inversion in *Arachnocampa flava* in Keroplatidae [11], and inversion and transposition of tRNA*s* in Sciaridae [25]. The conservative character, maternal inheritance, and straightforward gene orthologs of the mitogenome have made it a significant source for molecular markers to be applied in evolutionary and phylogenetic studies [26]. Phylogenetic studies based on insect mitogenomes have shown good results in Diptera [27], Orthoptera [28], Coleoptera [29], and Hymenoptera [30]. The knowledge of mitogenome characteristics in the infraorder Bibionomorpha is still limited, and mitogenomes of the vast majority of species within Bibionomorpha are still unknown or remain uncompleted, although the whole genomes of several species of Bibionomorpha have recently been published [31].

In this study, we report, for the first time, complete and annotated sequences of mt genomes of *P. simplioipes* and *P. hardyi* representing the family Bibionidae. We performed comparative analyses of complete mitogenomes of the 20 species from eight families of four superfamilies from infraorder Bibionomorpha. Additionally, we constructed the phylogenetic relationships and discussed the evolution of the infraorder based on hitherto-known mitogenome sequences. The study also provides a comprehensive information frame for further studies on the mitogenome of the Bibionomorpha.

## 2. Material and Methods

### 2.1. Sample Collecting and Mitogenome Sequencing

Samples of two species representing Bibionidae, *P. simplioipes* and *P. hardyi,* were collected from the Wuyi Mountains of Fujian Province, China in October 2018, and were preserved in individual vials in silica. After morphological identification using traditional method [32], they were stored in 100% alcohol, and housed at −20 °C until the DNA extraction. Total DNA was extracted from thoracic muscle tissues using the DNeasy tissue kit (Qiagen, Hilden, Germany) following the manufacturer’s instructions. Concentration of extracted genomic DNA was determined by Qubit 2.0 (Invitrogen, Shanghai, China). The genomic DNA library was constructed with 350 bp small fragment, and sequenced using Illumina HiSeq X sequencing technology in HuiTong Company at Shenzhen, China. The adapters and unpaired, short, and low-quality reads were removed from the raw dataset during the quality control by FastQC v 0.11.3 [33].

### 2.2. Mitogenome Assembly and Annotation

The clean reads of mitogenome were separated and assembled using de novo assembler SPAdes v 3.11.0 with default parameters using the known dipteran mitogenome sequences of *Clephydroneura* sp. and *Drosophila yakuba* as references [34]. The contigs of mitogenome were extracted and assembled into mitogenomes through searching against the reference sequences using PRICE (paired-read iterative contig extension) by NOVOPlasty version 2.6.2 [35]. The position and sequences of PCGs, tRNAs, rRNAs, and CR were originally annotated using the MITOS web server [36], and then determined in comparison with published homologous mitogenome sequences in phylogeny-close species using MEGAX [37]. The secondary structures of tRNAs were predicted by the MITOS, ARWEN 1.2, and tRNAscan-SE web Server [38]. The annotation of the mitogenomes was corrected manually using the Geneious v 4.8.5 [39], and final mitogenomes were submitted to the GenBank database.

### 2.3. Mitogenome Characteristics Analysis

The complete mitogenome for two newly sequenced Bibionoidea species were cyclized using the CGView online server with default parameters [40] (Table 1). Base composition, codon usage, relative synonymous codon usage (RSCU), and amino acid content of 20 species of mitogenome were computed with MEGA v.7.0.26 [41]. Nucleotide compositional bias was calculated using the following formulae: AT-skew = (A − T)/(A + T) and GC-skew = (G − C)/(G + C) by MEGA v.7.0.26, and three-dimensional scatterplots of AT-Skew, GC-Skew and AT% were drawn using Origin Pro v.9.0 [42]. Selection pressure of the 13 PCGs was analyzed by calculating Ka (non-synonymous) and Ks (synonymous substitution) values with DnaSP v6.12.03 [43], and visualized using Origin Pro v.9.0. The gene rearrangement events of tRNA*s* and PCG*s* were identified in comparison with the gene order of the ancestor Clephydroneura sp. using the Geneious v 4.8.5.

### 2.4. Phylogenetic Analysis

Phylogenetic analysis was conducted with both ML and BI using PhyloSuite v.1.2.2 [44]. Two datasets of mitogenomes were used for phylogenetic inference, and two species from the superfamily Tipuloidea, *Symplecta hybrida* and *Tipula cockerelliana* were employed as outgroup (Table 2). we constructed two nucleotide datasets: (1) PCG123, all codon positions, and (2) PCG123 + R, all codon positions + two rRNA genes. The nucleotide sequences of 13 PCGs were aligned by codon-based multiple alignments using the L-INS-i algorithm and the rRNAs were aligned using the Q-INS-i strategy in MAFFT [45]. The concatenation of aligned sequences was performed using SequenceMatrix [46]. The selection of best-fit partitioning schemes and substitution models for each dataset were calculated using PartitionFinder 2 with the following settings: branch lengths as linked, and model election as AICc with the greedy algorithm [47]. Partitioning schemes and models are listed in Appendix A. ML analysis was performed using IQ tree ver.1.6.8 under edge-linked partition model with 1000 bootstrapped replicates (BP) [48,49]. BI analysis was conducted using MrBayes ver. 3.26 [50] with two MCMC runs, each with four chains (three heated and one cold) run for 10,000,000 generations. Posterior probabilities (PPs) were computed after a 25% burn rate and each set was sampled every 1000 generations. An average deviation of the split frequency of less than 0.01 indicates that the runs had converged. The phylogenetic tree was visualized using FigTreev.1.4.4 and iTOL online tool [51,52].

### 2.5. Divergence Time Estimation

The divergence time was estimated using BEAST v.1.8.4 [53]. A γ distribution (GTR + I + G) nucleotide substitution model was selected by PartitionFinder 2.0 using AIC, and the speciation Yule model were selected as the tree priors with the uncorrelated lognormal relaxed molecular clock model. Two independent Markov chain Monte Carlo (MCMC) runs, each with a chain length of 100,000,000 generations with sampling every 1000 generations and a first 25% burn-in, were performed to estimate the divergence time. The fossil record of the species *Eoditomyia primitiva* 180 million years ago (Mya) for Sciaroidea was selected for calibration using the fossil calibration database and reported research searching in this study [54].

## 3. Results

### 3.1. Mitogenome Characteristics of Bibionomorpha

The complete mitogenomes of *P. simplioipes* (GenBank no.: MT511121) and *P. hardyi* (GenBank: MT511122) are both of circular, closed, and double-stranded structures, with full lengths of 15,349 and 15,456 bp, respectively (Figure 1). The complete mitogenomes of 20 species contained 37 genes (including 13 PCGs, 22 tRNA genes, and 2 rRNA genes) and *CR*. There are 22 genes (9 PCGs and 13 tRNAs) located on the majority coding strand (J-strand), while the other 15 genes (4 *PCG*s, 9 tRNAs, and 2 rRNAs) are on the minority strand (N-strand). The length of 20 mitogenome sequences ranges from 14,503 bp (*Rhopalomyia pomum*) to 16,950 bp (*Acnemia nitidicollis*), and the length variation mainly results from the *CR*, intergenic overlap, and spacers. The nucleotide base compositions and the three-dimensional scatter plot of the AT content, AT-skew, and GC-skew of 20 mitogenomes in the infraorder are shown in Appendix A and Figure 2. They all display obvious AT bias with A + T content ranging from 75.8% (*Allodia* sp.) to 85.7% (*Orseolia oryzae*). AT-skew values range from −0.034 (*Arachnocampa flava*) to 0.105 (*Orseolia oryzae*), and GC-skew from −0.326 (*Pnyxia scabiei*) to −0.102 (*Epicypta* sp.). Most of the species of the Bibionidae, Cecidomyiidae, and Sciaridae have similar AT content and AT/GC-skew, which are closely distributed in the three-dimensional scatter plot, whereas the species of the Mycetophilidae is widely distributed in the plot for AT content, AT-skew, and GC-skew. The total PCGs nucleotide length ranges from 10,794 bp (*Orseolia oryzae*) to 11,237 bp (*Sylvicola fenestralis*). Most of the *PCG*s initiate with the typical start codon ATN and TTG, whereas the special start codons TCA, TTG, and TCG are found for *cox1*; TTG for nad1; and GTG for *nad5*. The most frequently used stop codons are TAA and T, followed by the stop codons TAG and TA. The RSCU values and amino acid usage frequency of the 20 species of mitogenomes in the infraorder are presented in Appendix A and Figure 3. UUA is the most frequently used codon, followed by UCU, CGA, and GCU. Leu has the highest usage percentage for all mitogenomes investigated with an average of 16.46%, and Cys has the lowest percentage (0.66%). The usage percentages of amino acids reveal no obvious difference among different families. To evaluate the selective pressures of 13 PCGs of the infraorder, pairwise analyses of the non-synonymous (Ka) and synonymous (Ks) substitution ratio (Ka/Ks) are shown in Figure 4. The Ka/Ks ratios for all 13 genes are all less than 1, ranging from 0.03 in *cox1* to 0.98 in *atp8* in the following order: *cox1*, *cytb*, *cox3*, *cox2*, *atp6*, *nad5*, *nad3*, *nad1*, *nad4*, *nad2*, *nad4L*, *nad6,* and *atp8*. These results imply that all of these 13 *PCG*s experienced purifying selection. Most tRNAs can be folded into the typical clover-leaf structure containing four stems and loops except *trnS2*, in which the dihydrouracil (DHU) arm is absent. The lack of a DHU arm in this infraorder has been commonly observed across all of these 20 mitogenomes. *rrnL* is located between *trnL1* and *trnV*, and *rrnS* between *trnV* and *CR*. There are 12 mismatched base pairs (G-U) to be found in *P. simplioipes* tRNAs, and 13 mismatched base pairs (G-U) in *P. hardyi* (Appendix A).

### 3.2. Gene Rearrangement Events of Bibionomorpha

Compared with the putative ancestral mitogenome (e.g., *Drosophila yakuba*), there are no gene rearrangements to be found in the superfamilies Scatopsoidea, Anisopodoidea, and Bibionoidea, and the family Mycetophilidae in Sciaroidea (Figure 5); however, the arrangement events exist in some families in the Sciaroidea. In Keroplatidae, the *trnE* has been inverted in *Arachnocampa flava*. In Cecidomyiidae, the arrangement was observed in the genome upstream of *nad5* (13 tRNAs and 5 PCGs), and downstream of *nad5* with only one rearrangement (*trnT* and *trnP*). In *Rhopalomyia pomum*, three tRNAs (*trnE*, *trnT,* and *trnP*) have been inverted, *trnN* has been transposed from the position (*trnR*-*trnS1*) to a position (*trnG*-*nad3*), and *trnI* has been remote-inverted from the position (*rrnS*-*trnQ*) to a position (*trnF*-*trnS1*).

In *Mayetiola destructor*, three tRNAs (*trnE*, *trnT*, and *trnP*) have been inverted, two tRNAs (*trnN* and *trnY*) have been transposed from typical position to the position (*trnG*-*nad3*) and (*trnS1*-*trnE*), respectively, and *trnI* has been remote-inverted (i.e., translocation + inversion) from a typical position to the position *trnR*-*trnS1*. In *Orseolia oryzae*, ten tRNAs have been remote-inverted, and five tRNAs and five PCGs have been transposed. In Sciaridae, the *trnC* has been remote-inverted from the position (*trnW*-*trnY*) to a position (*trnQ*-*trnM*), and the *trnA* and *trnR* were switched in *Trichosia lengersdorfi*. The *trnC* has been remote-inverted from the position (*trnW*-*trnY*) to the position (*trnQ*-*trnM*), and the *trnY* and the *trnL2* have been remote-inverted, both of them from between *trnC* and *cox2* to between *rrnS* and *nad2* in *Pseudolycoriella* sp.

### 3.3. Phylogenetic Relationships and Divergence Time

Four phylogenetic trees were generated based on datasets PCG123 + R and PCG123 of 24 mitogenomes in Bibionomorpha. Two same topologies of trees from BI analysis (Figure 6) are slightly different from the two same trees from ML (Figure 7) in the positions *Arachnocampa flava* in Keroplatidae and *Pnyxia scabiei* in Sciaridae. The former is grouped with Cecidomyiidae in BI, but at the base of Cecidomyiidae + Sciaridae in ML, and the latter with *Trichosia lengersdorfi* in BI, but at the base of Sciaridae in ML.

The superfamily Scatopsoidea is located at the base of the infraorder Bibionomorpha, and the superfamily Anisopodoidea at the base of the superfamilies Bibionoidea + Sciaroidea. The Bibionoidea + Sciaroidea appears monophyletic with PP = 1 and BP = 100 (Figure 6 and Figure 7), and both of them are sister groups. In Bibionoidea, there are only three species to be investigated, and monophyly of the family Bibionidae cannot be determined. In Sciaroidea, there are four families Mycetophilidae, Keroplatidae, Cecidomyiidae, and Sciaridae to be investigated. The family Mycetophilidae seems monophyletic (PP = 1 and BP = 100), and is a sister group with Keroplatidae + Cecidomyiidae + Sciaridae (PP = 1 and BP = 100). The Sciaridae are monophyletic with PP = 1 and BP = 100. The position of Keroplatidae cannot be determined due to the position difference from both BI and ML analyses.

The PCG + R dataset was used to estimate divergence time because PCG + R and PCG resulted in the same topologies of phylogenetic trees using BI, but PCG + R had higher node support values than PCG in the initial phylogenetic assessment. We choose Sciaroidea as the fossil calibration point, and it was reported to originate at 180 Mya by Grimaldi and Engel. The Bibionomorpha was inferred to originate at 218 Mya, at least, in the Triassic (Figure 8). The superfamily Scatopsoidea is the earliest-appearing group in the investigation in the infraorder, and is estimated to originate at 218 Mya in the late Triassic. Subsequently, the Anisopodoidea appeared at 208 Mya in the late Triassic, and Bibionoidea + Sciaroidea at 190 Mya in the early Jurassic. The divergence of Bibionoidea might be tracked back to 149 Mya in the late Jurassic, and Sciaroidea to 180 Mya, in the early Jurassic. In Sciaroidea, the Mycetophilidae diverged at 150 Mya (in the late Jurassic), and the Keroplatidae at 170 Mya in the Jurassic. The Cecidomyiidae + Sciaridae was derived at 155 Mya in the late Jurassic, the Cecidomyiidae at 101 Mya, and Sciaridae at 134 Mya.

## 4. Discussion

### 4.1. Mitogenome Organization and Characteristics

The present study sequenced and annotated mitogenomes of two species in Bibionidae, which is the first report of complete mitogenomes for the superfamily Bibionoidea. There are 37 genes (13 PCGs, 22 tRNAs, and 2 rRNAs) and a *CR*, with high AT bias in nucleotide composition; the UAA, CGA, and GGA are three most used codons, and the Leu, Phe, and Ile are the most used amino acid in 20 species of Bibionomorpha mitogenomes investigated in this study. All tRNAs have a complete secondary structure except for *trnS2,* which lacks the DHU arm. All of these characteristics are identical to the reported mitogenomes of other dipterans [27,55]. The Ka/Ks ratios in 13 PCGs range from the 0.03 in *cox1* to the 0.98 in *atp8*, which implies that these PCGs have undergone purifying selection. This is consistent with earlier studies in Culicidae [55,56], and in Mycetophilidae [57].

### 4.2. Gene Rearrangement of Mitogenomes

There are some rearrangement events in a small number of species in the superfamily Sciaroidea, but not in analyzed representatives of Scatopsoidea, Anisopodoidea, and Bibionoidea. The rearrangements exist in the families Keroplatidae, Sciaridae, and Cecidomyiidae but not in Mycetophilidae in the Sciaroidea. The unique inversion in *Arachnocampa flava* in Keroplatidae has been reported in other dipteran species, such as *Anopheles quadrimaculatus* and *An. gambiae* in Culicidae [58,59]. These three inversions, seven transpositions, and fourteen inverse transpositions (remote inversion) in three species in Cecidomyiidae have also been reported in *Paracladura trichoptera* in Trichoceridae in Diptera [11]. These two inverse transpositions and three transpositions in *Trichosia lengersdorfi* and *Pseudolycoriella* sp. in Sciaridae confirm earlier reports in Sciaridae [25].

Our phylogenetic study shows that the Sciaroidea is located at the top of four superfamilies investigated in this study, and Mycetophilidae and Keroplatidae + Cecidomyiidae + Sciaridae are sister groups in the Sciaroidea. These rearrangements exist only in the clade Keroplatidae + Cecidomyiidae + Sciaridae, which suggests that these rearrangement events appeared in the late period in the evolution of the Bibionomorpha. Their evolutionary mode and dynamics need to be explored with more mitogenome studies.

### 4.3. Phylogenetics and Evolution

The present study suggests the phylogenetic relationships of Scatopsoidea + (Anisopodoidea + (Bibionoidea + Sciaroidea)) in Bibionomorpha. Earlier molecular-based studies proposed Anisopodoidea to be thet most primitive group in Bibionomorpha [8,10], and another molecular-based study even suggested that the Anisopodoidea be included in one other infraorder Ptychopteromorpha [11]. Our result shows that the superfamily is derived later than Scatopsoidea, but only one species is investigated for both Anisopodoidea and Scatopsoidea in our study. Therefore, the phylogenetic position and monophyly of these two superfamilies are not yet settled. Although based on a very limited taxon sampling, this study supports the earlier research results for the monophyly of Sciaroidea [10,15]. The monophyly of Bibionidae in Bibionoidea is not yet determined due to only two species being included. The present study supports the monophyly of Mycetophilidae and Sciaridae in Sciaroidea, which was reported also based on mitogenomes [25,57]. However, further work and broader taxon sampling are necessary.

The present study suggests the origin of Bibionomorpha in the Triassic, which is consistent with an earlier study on Diptera evolution [60,61,62,63]. During the Triassic period, there were many autodigested fungi and decaying matter, which provided food for early lineages of the infraorder. The superfamilies Scatopsoidea and Anisopodoidea are suggested to be derived in the late Triassic and be the most primitive groups in the infraorder, which is consistent with the results of the previous study [9]. The Bibionoidea diverged in the Upper Jurassic (149 Mya), which is consistent with an earlier study [64]. Keroplatidae dates back to the Jurassic (170 Mya); the inferred dates are earlier than the inference by Blagoderov (105.3–99.7 Mya) [65]. The Mycetophilidae (150 Mya) and Cecidomyiidae + Sciaridae (155 Mya) in Sciaroidea are proposed to be derived in the Jurassic for the first time. The families Sciaridae (134 Mya) and Cecidomyiidae (101 Mya) in Sciaroidea originated in the Cretaceous, which is consistent with anearlier study [8]. In further research, more accurate estimates of divergence times are necessary with more precise fossil records for calibration and more complete sampling.

## 5. Conclusions

This is the first comprehensive study on the general characteristics of mitogenomes and mitogenomes-based phylogenetics in Bibionomorpha. Two species of mitogenomes are newly sequenced and annotated, and they represent the first report of complete mitogenomes for the family Bibionidae. A total of 20 mitogenomes in Bibionomorpha have been analyzed in the present study, and they are of the same general characteristics as other Diptera insects. Our results show that there are no gene rearrangements found in the superfamilies Scatopsoidea, Anisopodoidea, and Bibionoidea, and the family Mycetophilidae in Sciaroidea, but gene rearrangements exist in some families (Keroplatidae, Sciaridae, and Cecidomyiidae) in the Sciaroidea, which suggests that these rearrangement events are derived in the late period in the evolution of the Bibionomorpha. Our results support the phylogenetic relationships of Scatopsoidea + (Anisopodoidea + (Bibionoidea + Sciaroidea)) in Bibionomorpha, and the monophyly of the families Sciaridae and Mycetophilidae. In addition, we inferred that the Bibionomorpha originated in the Triassic, Scatopsoidea and Anisopodoidea in the late Triassic, Bibionoidea in the Jurassic, and Sciaroidea in the Jurassic to the Cretaceous. Our results demonstrate that the mitogenomes can be used to infer phylogenetic relationships in Bibionomorpha to help understand the history and evolutionary biology of the infraorder Bibionomorpha.

## Figures and Tables

**Figure 1 genes-14-01485-f001:**
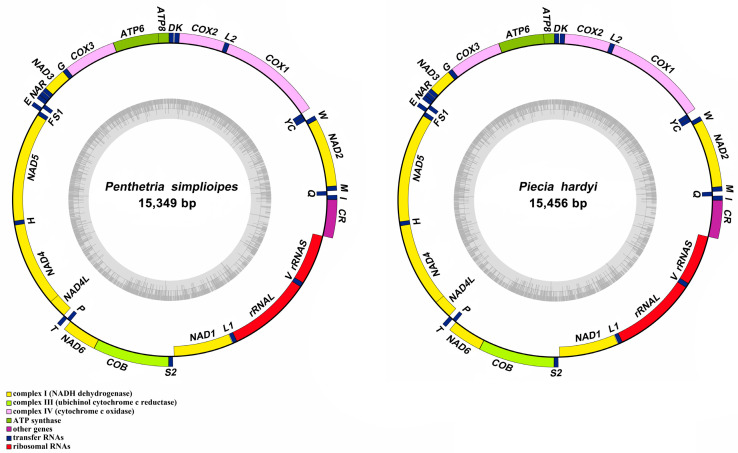
Mitogenome structure of *P. simplioipes* and *P. hardyi* newly sequenced and annotated in this study. The yellow, blue, red, and purple blocks indicate PCGs, tRNAs, rRNAs, and *CR*, respectively. The genes on the outer circle are located on the J-strand, whereas the genes on the inner circle are located on the N-strand. L1, L2, S1, and S2 represent the tRNAs tRNA-Leu (CUN), tRNA-Leu (UUR), tRNA-Ser (AGN), and tRNA-Ser (UCN), respectively.

**Figure 2 genes-14-01485-f002:**
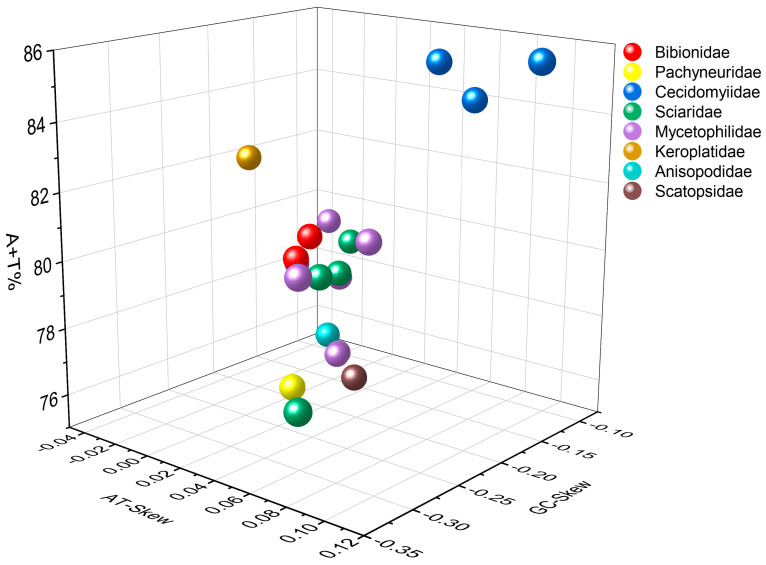
Three-dimensional scatter plot charts of the AT-Skew, GC-Skew, and AT content from 20 mitogenome sequences in the Bibionomorpha. The different colors of balls represent the different species.

**Figure 3 genes-14-01485-f003:**
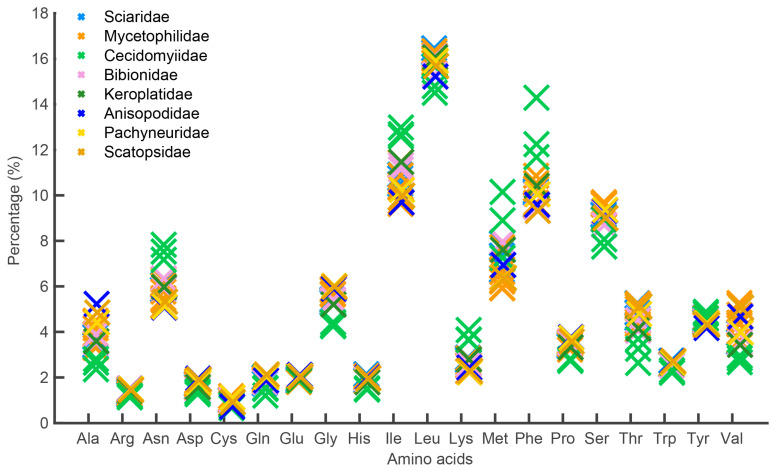
Frequency percentage of each of 20 coded amino acids in 20 mitogenome sequences in the Bibionomorpha.

**Figure 4 genes-14-01485-f004:**
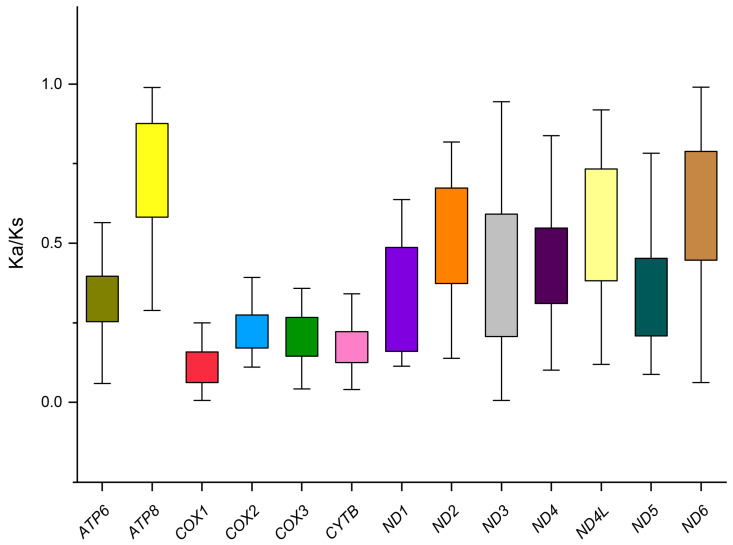
Evolutionary rates of 13 protein-coding genes (PCGs) within 20 mitogenomes in the Bibionomorpha. Ks: The number of synonymous substitutions per synonymous site; Ka: The number of non-synonymous substitutions per non-synonymous site; Ka/Ks: The ratio of non-synonymous substitutions to synonymous substitution.

**Figure 5 genes-14-01485-f005:**
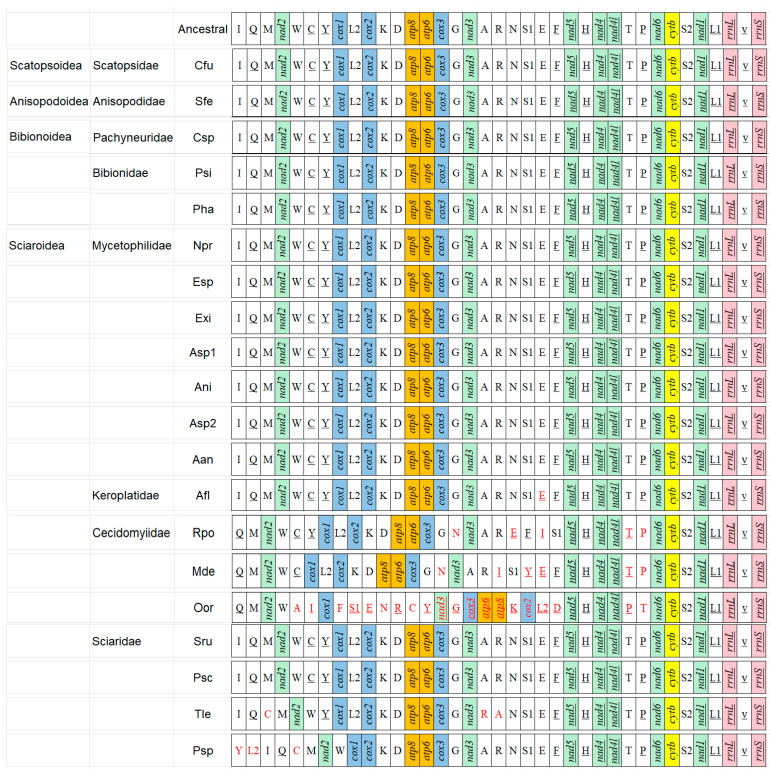
Mitogenome organization of the Bibionomorpha referenced with the ancestral insect mitogenomes. The underlined symbols are located on the N-strand and others on the J-strand. The white, green, blue, orange, yellow, and pink blocks denote transfer tRNAs, DNAH dehydrogenase, Cytochrome C oxidase, ATP synthetase, and ribosomal RNAs, respectively, with the red gene representing the existence of rearrangement. Abbreviations: Cfu: *Coboldia fuscipes*, Sfe: *Sylvicola fenestralis*, Csp: *Cramptonomyia spenceri*, Psi: *Penthetria simplioipes*, Pha: *Plecia hardyi*, Npr: *Neoempheria proxima*, Esp: *Epicypta* sp., Exi: *Epicypta xiphothorna*, Asp1: *Azana* sp., Ani: *Acnemia nitidicollis*, Asp2: *Allodia* sp., Aan: *Allodia anglofennica*, Afl: *Arachnocampa flava*, Rpo: *Rhopalomyia pomum*, Mde: *Mayetiola destructor*, Oor: *Orseolia oryzae*, Sru: *Sciara ruficauda*, Psc: *Pnyxia scabiei*, Tle: *Trichosia lengersdorfi*, Psp: *Pseudolycoriella* sp.

**Figure 6 genes-14-01485-f006:**
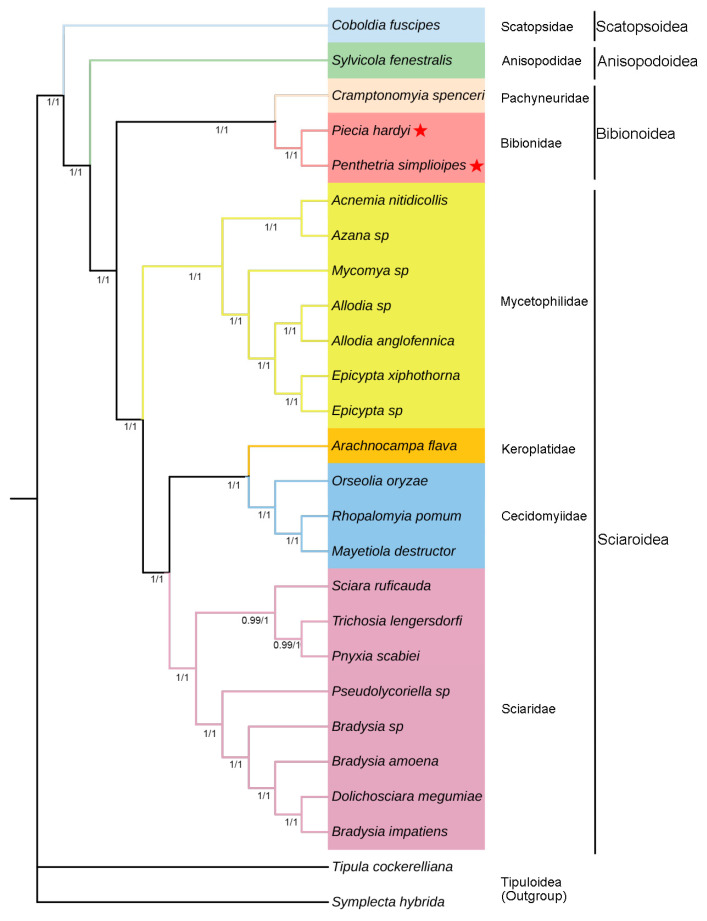
Mitogenome-based phylogenetic relationships of Bibionomorpha species. They are constructed based on PCG123 + R datasets and PCG123 datasets using BI method. The posterior probabilities PCG123 + R are shown on corresponding nodes in the identical topology of BI tree, with left and right values separately based on PCG123 + R and PCG123 datasets. The different colors of species name blocks represent the different families. The mitogenomes of two species newly sequenced in this study are indicated by pentagrams.

**Figure 7 genes-14-01485-f007:**
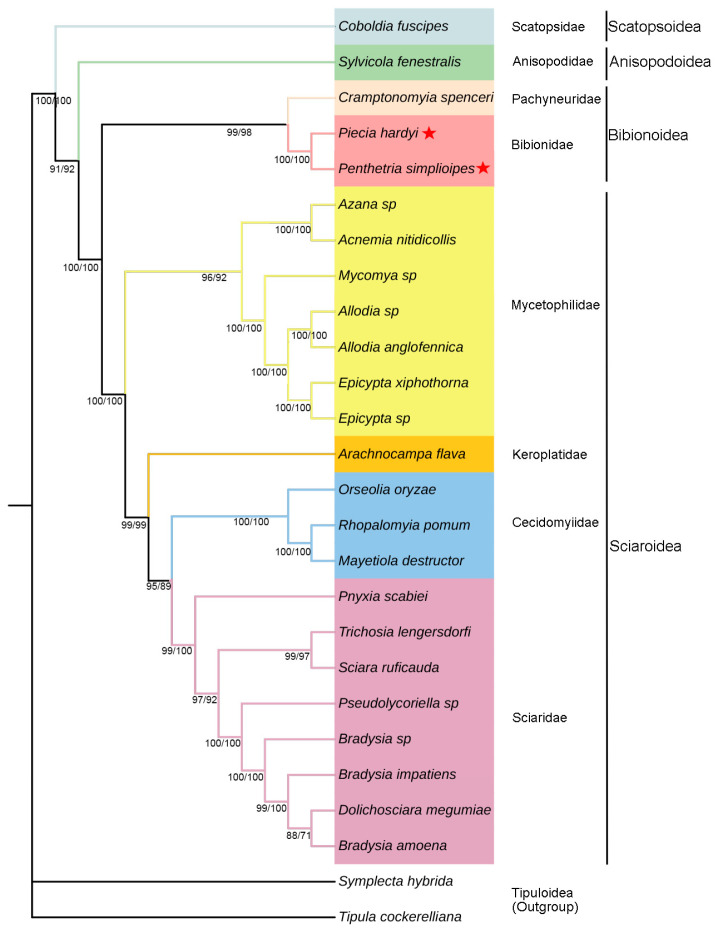
Mitogenome-based phylogenetic relationships of Bibionomorpha species. They are constructed based on PCG123 + R and PCG123 datasets using ML method. The bootstrap values PCG123 + R are shown on corresponding nodes in the identical topology of ML tree, with left and right values separately based on PCG123 + R and PCG123 datasets. The different colors of species name blocks represent the different families. The mitogenomes of two species newly sequenced in this study are indicated by pentagrams.

**Figure 8 genes-14-01485-f008:**
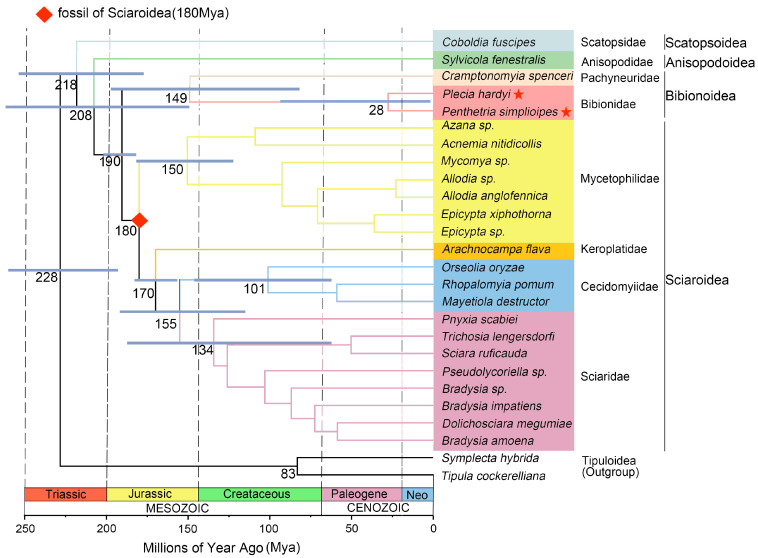
The divergence time produced by the BEAST analysis of the PCG + R dataset using one fossil calibration point, which is indicated by rectangle. Blue bars represent 95% credibility intervals for the ages of the clades. The mitogenomes of two species newly sequenced in this study are indicated by pentagrams. A geological time scale is shown at the bottom.

**Table 1 genes-14-01485-t001:** Organization of the mitogenomes of *Penthetria simplioipes* (PSI) and *Plecia hardyi* (PHA) newly sequenced in this study.

Gene	Strand	Position (bp)		Length (bp)		Space (+)/Overlap (−)		Start/Stop Codon	
		PSI	PHA	PSI	PHA	PSI	PHA	PSI	PHA
*trnI*	J	1–67	1–71	67	71	0	0		
*trnQ*	N	84–152	68–138	69	71	16	−4		
*trnM*	J	211–279	136–204	69	69	58	−3		
*nad2*	J	279–1325	204–1244	1047	1041	−1	−1	ATA/TAA	ATT/TAA
*trnW*	J	1326–1395	1266–1334	70	69	0	21		
*trnC*	N	1394–1459	1327–1390	66	64	−2	−8		
*trnY*	N	1465–1533	1398–1466	69	69	5	7		
*cox1*	J	1568–3065	1501–2998	1498	1498	34	34	ATT/T	ATC/T
*trnL1*	J	3065–3132	2998–3065	68	68	−1	−1		
*cox2*	J	3137–3820	3069–3761	684	693	4	3	ATT/TAA	ATG/TAA
*trnK*	J	3824–3896	3761–3831	73	71	3	−1		
*trnD*	J	3903–3974	3837–3905	72	69	6	5		
*atp8*	J	3974–4132	3906–4067	159	162	−1	0	ATT/TAA	ATT/TAA
*atp6*	J	4126–4803	4061–4735	678	675	−7	−7	ATG/TAA	ATG/TAA
*cox3*	J	4805–5593	4739–5527	789	789	1	3	ATG/TAA	ATG/TAA
*trnG*	J	5594–5660	5528–5595	67	68	0	0		
*nad3*	J	5661–6014	5596–5947	354	352	0	0	ATT/TAA	ATT/TAA
*trnR*	J	6020–6085	5947–6011	66	65	5	−1		
*trnA*	J	6083–6149	6011–6079	67	69	−3	−1		
*trnN*	J	6149–6218	6079–6147	70	69	−1	−1		
*trnS2*	N	6218–6286	6152–6207	69	56	−1	4		
*trnE*	J	6286–6356	6216–6281	71	66	−1	8		
*trnF*	N	6355–6421	6281–6352	67	72	−2	−1		
*nad5*	N	6421–8152	6352–8083	1732	1732	−1	−1	ATT/T	ATT/T
*trnH*	N	8153–8219	8084–8148	67	65	0	0		
*nad4*	N	8219–9554	8149–9489	1336	1341	−1	0	ATT/TAA	ATG/TAA
*nad4L*	N	9548–9844	9480–9776	297	297	−7	−10	ATG/TAA	ATG/TAA
*trnT*	J	9852–9914	9785–9849	63	65	7	8		
*trnP*	N	9914–9981	9849–9916	68	68	−1	−1		
*nad6*	J	9983–10,501	9918–10,436	519	519	1	1	ATT/TAA	ATA/TAA
*cytb*	J	10,501–11,637	10,445–11,581	1137	1137	−1	8	ATG/TAA	ATG/TAA
*trnS1*	J	11,641–11,712	11,584–11,654	72	71	3	2		
*nad1*	N	11,726–12,658	11,672–12,628	933	957	13	17	TTG/TAG	ATT/TAG
*trnL2*	N	12,677–12,745	12,629–12,700	69	72	18	0		
*rrnL*	N	12,746–13,752	12,701–14,050	1007	1350	0	0		
*trnV*	N	13,753–13,826	14,051–14,124	74	74	0	0		
*rrnS*	N	13,827–14,611	14,125–14,909	785	785	0	0		
CR		14,612–15,349	14,910–15,465	738	556	0	0		

**Table 2 genes-14-01485-t002:** Taxonomic and mitogenome information of species used in the present phylogenetic analysis.

Superfamily	Species	Total	PCGs	tRNA	rRNA	CR	GenBank
/Family		Size (bp)	Size (bp)	Size (bp)	Size (bp)	Size (bp)	ID
Bibionoidea							
Bibionidae	*Penthetria simplioipes*	15,349	11,163	1513	1792	738	MT511121
	*Piecia hardyi*	15,456	11,193	1493	2084	556	MT511122
Pachyneuridae	*Cramptonomyia spenceri*	16,274	11,223	1477	2138	1068	NC016203
Scatopsoidea							
Scatopsidae	*Coboldia fuscipes*	15,309	11,167	1470	2111	510	MZ567016
Anisopodoidea							
Anisopodidae	*Sylvicola fenestralis*	16,234	11,237	1432	2133	1232	NC016176
Sciaroidea							
Cecidomyiidae	*Mayetiola destructor*	14,759	11,209	1474	2048	604	NC013066
	*Orseoli aoryzae*	15,286	10,794	1474	1988	578	NC027680
	*Rhopalomyia pomum*	14,503	10,897	1477	1995	363	GQ387649
Sciaridae	*Sciara ruficauda*	15,167	11,204	1476	2116	N/A	MN161586
	*Bradysia* sp.	15,512	11,124	1377	2130	N/A	MN161585
	*Bradysia amoena*	14,049	10,032	1484	1962	N/A	NC057971
	*Bradysia impatiens*	16,479	11,217	1450	2185	370	MZ202360
	*Pnyxia scabiei*	15,437	11,215	1463	2128	N/A	NC053636
	*Trichosia lengersdorfi*	16,171	11,222	1477	2212	N/A	MN161589
	*Pseudolycoriella* sp.	15,981	11,224	1482	2163	N/A	MN161587
	*Dolichosciara megumiae*	15,931	11,114	1423	2132	N/A	MN161588
Mycetophilidae	*Neoempheria proxima*	15,853	11,205	1469	2112	783	MW116802
	*Epicypta* sp.	16,279	11,199	1457	2116	6145	MW116800
	*Epicypta xiphothorna*	15,902	11,196	1452	2132	563	MW116799
	*Azana* sp.	16,654	11,226	1507	2171	762	MW116798
	*Acnemia nitidicollis*	16,950	11,206	1478	2132	N/A	NC050318
	*Allodia* sp.	14,899	11,199	1465	2077	N/A	MN310892
	*Allodia anglofennica*	14,897	11,196	1467	2074	N/A	MN310891
Keroplatidae	*Arachnocampa flava*	16,923	11,202	1482	2103	1841	NC016204
Outgroup (Tipulomorpha)							
Tipulidae	*Symplecta hybrida*	15,811	11,167	1456	2109	897	NC030519
	*Tipula cockerelliana*	14,453	11,154	1378	1837	N/A	NC030520

## Data Availability

The GenBank number of two newly sequenced species *P. simplioipes*: MT511121, *P. hardyi*: MT511122.

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
