# Peer review of "Two New Mitogenomes of Bibionidae and Their Comparison within the Infraorder Bibionomorpha (Diptera)"

_genes, 2023, doi:10.3390/genes14071485_

Round 1

Reviewer 1 Report

I am sending my comments in the attached file.

Author Response

Dear reviewer,

Thank you for your valuable comments, we are sorry for our carelessness. Based on your comments, we have made extensive corrections to our previous draft. The more details presented in the revised manuscript.

Would you check if it is acceptable?

Best regards.

Bin Chen

Reviewer 2 Report

The subject of the study is interesting. The study was performed at a very high scientific and methodological level. I am very pleased that the authors use a combination of traditional and modern approaches for the species identification.

It is written in a clear language.

Is it possible to increase the resolution of the figures used, some text is poorly readable.

Line 55 – at the first mention, specify what is BI? This is referred to for the first time on lines 161-162.

Figure 8. It would be interesting to see the deviation levels on the divergence time graph.

Author Response

Dear reviewer,

Thank you for your comments, we have made extensive corrections to our previous draft and tried our best to polish the language in the revised manuscript. Figure 8 has been modified follow your suggestion. The more details presented in the manuscript.

Would you check if it is acceptable?

Best regards.

Bin Chen

Reviewer 3 Report

Major issues

Literature cited: The authors are not specialists on the taxonomy and systematics of Bibionomorpha, so that some important papers on the phylogeny of Bibionoidea and Sciaroidea are not discussed in the text. The authors should consult at least some of the following papers focused on the phylogeny of various groups of the infraorder: Chandler (2002), Jaschhof (2011), Hippa & Vilkamaa (2005, 2006), Dorchin et al. (2019), Sikora et al. (2019), Kasprak et al. (2019), Mantic et al. (2020), etc. Representatives of Bibionomorpha are also very common in the fossil record, so some of the recent papers on this topic should also be discussed (e.g. Grimaldi et al. 2003, Skartveit 2008, Sevcik et al. 2021, Skartveit 2023).

Taxon sampling: The remarkably poor, insufficient and unbalanced taxon sampling is one of the most serious flaws of this manuscript. Taxon sampling is uneven either taxonomically or geographically. Representatives of six extant families of Bibionomorpha sensu lato are missing in the dataset: Axymyidae, Bolitophilidae, Diadocidiidae, Ditomyiidae, Hesperinidae, and Rangomaramidae. No species of Sciaroidea incertae sedis is included. The phylogenetically important and diverse family Keroplatidae is represented only by a single species of the very specific genus Arachnocampa. The heterogeneous and megadiverse family Cecidomyiidae is represented only by 3 species from a single (and most advanced) subfamily. No phylogenetically really important or problematic taxa are included in the dataset (except Arachnocampa) and taxon sampling is restricted to Northern Hemisphere, although many important taxa occur exclusively in Southern Hemisphere.

Actually, we now live in the era of next-generation sequencing, with new whole genome and transcriptome data being published every year. There are, for example, whole genomes of several taxa missing in the dataset published by Anderson et al. (2022, PNAS) and available on GenBank. The authors should try to reconstruct mitogenomes from these data, to improve their taxon sampling.  

On the other hand, two or three species of the same genus are included in the dataset (Alodia, Bradysia, Epicypta), without any explanation or logic. Such a taxon sampling thus makes no sense and it is difficult to talk about a reconstruction of phylogeny but rather about preliminary attempt to compare several mitogenomes, without ambition to solve any important phylogenetic hypothesis.

The new mitogenome sequences should be available on GenBank as soon as the paper is published. For the species taken from GenBank (majority of species in the dataset), please provide references to original publications.

Phylogenetic analysis: The phylogenetic trees presented are based on very incomplete and unbalanced selection of taxa (see above). Such phylogenetic trees may be rather misleading than helpful. It is actually a comparison of two new mitogenomes with several others from the GenBank database than a real phylogeny of such a huge group. From this point of view, the title of the paper is very inflated and should be modified, see below. My suggestion is either to remove the “phylogenetic part” from the manuscript or re-formulate and shorten these parts accordingly, especially the sentences which include any mention about the monophyly of the families in question should be deleted. No monophyly can be established using such a small taxon sampling. Also the part devoted to divergence time estimation makes little sense, from the same reason.

Minor issues

There are numerous errors in the Latin names of the taxa, especially in Tables 2, S2, S3, but also at many places in the text. Please double-check and correct everything carefully.

Latin names of genera and species should be written in italics, please correct everywhere throughout the text.

The title of the paper should be modified, e.g. to Two new mitogenomes of Bibionidae and their comparison within the infraorder Bibionomorpha (Diptera)

The paraphyletic taxon Nematocera is no longer used and should be replaced by “lower Diptera” or or with mentioning of particular infraorders.

The genera Penthetria and Plecia are currently considered as members of the family Bibionidae. Using the name Bibionoidea by the authors is thus unnecessary.

The family Lygistorrhinidae is no longer considered as a separate family (cf. line 53) but only as a subfamily of Keroplatidae (see Mantic et al. 2020).

Line 67 – correct the paragraph.

Line 180 – missing full stop.

Line 182 – correct the word mtgenome. I would prefer using mitogenome throughout the text, rather than mtgenome.

Author Response

Dear reviewer,

Thank you for your comments, we have made extensive corrections to our previous draft. About three major issues, I have the following answer. First, we have checked the literature carefully and added more references; second, about insufficient and unbalanced taxon sampling, we have searched for the GenBank and no more complete Mitochondrial genome as supplementary data, so the data volume has not changed; Finally, based on your further suggestions, “phylogenetic part” has been made shorten. About minor issues, we I have checked and corrected the Latin names of the taxa throughout the manuscript, the title has been modified follow your suggestion and other issues have also been modified. The more details presented in the manuscript.

Would you check if it is acceptable?

Best regards.

Bin Chen

Round 2

Reviewer 3 Report

The text still contains many errors and incorrect formulations. Some of the claims related to the phylogeny of the infraorder are still too strong considering very low taxon sampling.

The authors should read their text carefully and correct at least these issues:

Line 11: remove “far unsolved” and insert “controversial”

Line 12: not” Penthetria simplioipes and Plecia hardyi” but “Penthetria simplioipes Brunetti, 1925 and Plecia hardyi Yang & Luo, 1988”

Line 13: not “Bibionidae, this is” but “Bibionidae. This is”

Line 15: not “Infraorder” but “infraorder”

Lines 23–25: correct the sentence to: “The work lays a base for the study of mitogenomes in Bibionomorpha but further work and broader taxon sampling is necessary for better understanding of phylogenetics and evolution of the infraorder.”

Line 39: Bradysia should be in italics

Line 43: not “The classification of the Bibionomorpha started more than 200 years ago” but rather “The classification of the Bibionomorpha has been studied almost 200 years”

Line 66: not “Manti” but “Mantič”

Line 68: not “Cecidomyiidae was the monophyly” but “Cecidomyiidae was monophyletic” – here the comprehensive paper about Cecidomyiidae phylogeny by Sikora et al (2019) in Zool. J. Linn. Soc. should be consulted and cited.

Line 98: not “and a comprehensive mitogenome-based phylogenetic analysis of Bibionomorpha can further reveals the phylogeny and evolution of the infraorder” but “mitogenomes of the vast majority of species within Bibionomorpha are still unknown or remain uncompleted, although the whole genomes of several species of Bibionomorpha have recently been published” – here the study of Anderson et al. (2022) in PNAS (https://doi.org/10.1073/pnas.2122580119) can be cited, which brings 14 whole genomes of representatives of various families of Bibionomorpha.

Line 101: Not “we” but “We”

Line 103: not “What’s more,” but “Additionally,”

Line 105: not “we investigated” but “hitherto known”

Line 154: not “hybrid” but “hybrida”

Table 2: not “Epicyptaxi phothorna” but “Epicypta xiphothorna”, not “Tipulaco ckerelliana” but “Tipula cockerelliana”

Line 181: not “primitive” but “primitiva”

Line 186: not “Mtgenoem” but “Mitogenome”

Line 304: not “the family Bibionidae is monophyly can’t be determined” but “monophyly of the family Bibionidae cannot be determined”

Line 331: remove “

Line 350: not “trichopteran” but “trichoptera”

Line 368: not “The present study supports” but “Although based on a very limited taxon sampling, this study supports”

Line 369: not “[15,10]” but “[10,15]”

Line 369: delete “and explore the relationship of these sister groups for the first time”

Line 372: delete “and reveals that Mycetophilidae is sister with Keroplatidae + Cecidomyiidae + Sciaridae for the first time” – this is only an artefact of the low taxon sampling

Line 373: delete “present study lays a base for the phylogenetics of the infraorder, and further work is necessary with more species involved” and insert only “However, further work and broader taxon sampling is necessary.

Line 453: not “Manti” but “Mantič”. Please include all the authors of this paper.

Double-check all the entries in References chapter.

Author Response

Dear reviewer,

We sincerely thank you for careful reading, and we were really sorry for our careless mistakes.  Based on your comments, we have carefully revised the manuscript, and are submitting its revised version.

Would you check if it is acceptable?

Best regards.

Bin Chen
